# The Single and Combined Effects of Prenatal Nonchemical Stressors and Lead Exposure on Neurodevelopmental Outcomes in Toddlers: Results from the CCREOH Environmental Epidemiologic Study in Suriname

**DOI:** 10.3390/children10020287

**Published:** 2023-02-02

**Authors:** Aloysius Ph. Koendjbiharie, Ashna D. Hindori-Mohangoo, Wilco C. W. R. Zijlmans, Jeffrey K. Wickliffe, Arti Shankar, Hannah H. Covert, Maureen Y. Lichtveld, Antoon W. Grünberg, Stacy S. Drury

**Affiliations:** 1Community Health Department, Regional Health Services, Paramaribo, Suriname; 2Faculty of Medical Science, Anton De Kom University, Paramaribo, Suriname; 3Foundation for Perinatal Interventions and Research in Suriname (Perisur), Paramaribo, Suriname; 4Department of Environmental Health Sciences, School of Public Health and Tropical Medicine, Tulane University, New Orleans, LA 70112, USA; 5Department of Environmental Health Sciences, School of Public Health, University of Alabama at Birmingham, Birmingham, AL 35294, USA; 6Department of Biostatistics and Data Science, School of Public Health and Tropical Medicine, Tulane University, New Orleans, LA 70112, USA; 7Department of Environmental and Occupational Health, School of Public Health, University of Pittsburgh, Pittsburgh, PA 15261, USA; 8Department of Psychiatry, School of Medicine, Tulane University, New Orleans, LA 70112, USA

**Keywords:** prenatal, lead, nonchemical stressors, neurodevelopment, toddlers, Suriname

## Abstract

The primary aim of this prospective study was to examine the single and combined effect of prenatal exposure to perceived stress, probable depression, and lead on toddlers’ neurodevelopment using the Bayley Scales of Infant and Toddler Development, third edition. Data from 363 mother-toddler pairs enrolled in the Caribbean Consortium for Research in Environmental and Occupational Health prospective cohort study were analyzed. A prenatal lead exposure of ≥3.5 µg/dL was associated with significantly lower receptive (*p* = 0.008) and expressive (*p* = 0.006) communication scaled scores. Moderate and severe maternal prenatal probable depression scores were associated with significantly lower fine (*p* = 0.009) and gross (*p* = 0.009) motor scaled scores. However, a maternal report of prenatal stress was not associated with neurodevelopmental outcomes. After adjusting for maternal demographics, prenatal stress and lead exposure, prenatal probable depression remained predictive of the toddlers’ gross motor scaled scores (β −0.13, 95% CI [−0.24–−0.02]). Similarly, when adjusting for demographics, prenatal stress and probable depression, prenatal lead exposure remained a significant predictor of their receptive communication scaled scores (β −0.26, 95% CI [−0.49–−0.02]). An analysis testing combined exposure to perceived stress, probable depression, and lead exposure, measured using a cumulative risk index, significantly predicted the child fine motor scaled scores after adjusting for other covariates (β −0.74, 95% CI: [−1.41–−0.01]).

## 1. Introduction

In the USA, the cases of neurodevelopmental disorders (NDDs), such as attention-deficit hyperactivity disorder, autism spectrum disorder, learning disabilities, and intellectual disabilities, among children between 3 and 17 years of age have increased in the last decade [1] and affect 10–15% of all births [2]. The rates of NDDs in low- and middle-income countries (LMICs), while less directly evaluated, are reported to be similar or higher than the rates in the United States, and it has been reported that perinatal complications and exposures are considered a predominant risk factor [3]. Estimations show that, globally, in 2016, 52.9 million children younger than five had developmental disabilities and that most of these children (95%) lived in LMICs [3]. Studies reported a pooled prevalence, globally, of 7.6 cases/1000 people under 19 years of age with the highest pooled prevalence being in Latin America: 33.4 cases/1000 [4]. NDDs represent a global public health concern reducing educational and vocational opportunities, lifetime productivity, and quality of life, resulting in substantial economic burdens on many societies [5,6]. Limited resources in LMICs decrease the likelihood that these children will reach their full developmental potential. 

The in utero environment plays a critical role in the trajectory of health risks for individuals across their life span. Due to its rapid rate of development, the fetal brain is particularly vulnerable to harmful exposure [7]. Adverse prenatal environments as the result of exposure to nonchemical stressors (NCSs), such as maternal psychosocial stress, as well as chemical stressors (CSs), such as lead exposure, are risk factors associated with alterations in the morphologic development of the fetal brain [8] and contribute to the elevated risk of NDDs. 

Maternal prenatal psychosocial stress is characterized by mental disorders such as perceived stress, anxiety, and depression [9,10,11]. Evidence indicates that prenatal psychosocial stress, maternal prenatal perceived stress, and depression, referred to as NCSs in this study, are each associated with adverse neurodevelopment in children. Studies on children of 14 to 19 months of age reported that prenatal exposure to NCSs is associated with lower Mental Developmental Index scores based on the second edition of the Bayley Scales of Infant and Toddler Development (BSID II) [12,13]. Tamayo Y Ortiz et al. [14] found an inverse association between the cognitive and language Bayley scores based on the third edition of the BSID (BSID III) for prenatal maternal stress exposure in 24-month-old children. In another study, Thomas et al. [15] reported that, compared with the toddlers of women who were not depressed, the 30-month-old toddlers of women with prenatal depression scored lower on the receptive language domain of the BSID III. 

Lead exposure remains a public health problem despite global efforts to reduce environmental lead levels [16]. The blood lead reference value (BLRV) in children and pregnant/lactating women in 2012 was 5 μg/dL [17,18] but was further reduced, in October 2021, to 3.5 µg/dL by the US Centers for Disease Control and Prevention (USCDC) [19]. Studies, however, indicate that there is no “safe” blood lead level and that any level of lead exposure prenatally is potentially detrimental to neurodevelopment [20,21]. Studies using the first version of the Bayley Scales (BSID I) to measure neurodevelopment in children from 12 to 24 months of age have reported an association between prenatal lead exposure and a lower Mental Developmental Index score [22,23,24]. Tamayo Y Ortiz et al. [14] found an inverse association between the cognitive and language BSID III scores for prenatal lead exposure in 24-month-old children. 

A low socioeconomic status and socially disadvantaged environments are not only risk factors for NCSs, but they also increase the risk of CSs in pregnant women and the developing fetus [25,26,27]. Therefore, when considering the adverse neurodevelopmental effects of prenatal maternal lead exposure, a better understanding of the maternal psychosocial environment in which the exposure is occurring is essential. While the independent association between NCSs as well as CSs and child neurodevelopmental outcomes has a significant literature base, less research has examined the combined or cumulative effects of the two sets of stressors. Combined exposure is of particular interest because of its potential cumulative and interactive effects on the neurodevelopment of the fetus and the possibility of co-occurrence [8,28,29]. Studies on prenatally exposed rodents provide evidence that NCSs and CSs interact and accentuate neurobehavioral toxicity through the shared targeting of neurological pathways such as the hypothalamic–pituitary–adrenal axis and the mesocorticolimbic dopamine (MCL-DA) system [8,25,26,28]. The few epidemiological studies that have focused on the combined effects of prenatal NCSs and lead exposure in human populations reported that they have interactive effects on neurodevelopmental indices in toddlers [14,30]. For example, Tamayo and colleagues [14] confirmed that prenatal stress and lead exposure were independently negatively associated with the scores in the BSID III with modest evidence of an interaction wherein increased prenatal stress potentiated the neurotoxicity of lead. 

While there is evidence of the health impacts of prenatal NCSs and CSs on mothers and children in general, there are no other published reports, to our knowledge, that measure the prevalence of NDDs in toddlers in Suriname, nor are there reports that evaluate the association between NDDs, NCSs, and lead exposure. Currently, there is no systematic screening for maternal prenatal NCSs in the primary care system, where most of the pregnant women in Suriname have their first prenatal consultations. Furthermore, there are no policies for monitoring the lead levels in the environment nor programs for the primary and secondary prevention of prenatal lead exposure. Suriname, to date, has not banned lead paint and only recently, in 2001, “phased” out leaded gasoline, long after most countries had done so. 

The lack of data on the epidemiology of NDDs in Suriname and the absence of systematic screening and prevention policies highlight the need for a careful assessment of prenatal maternal NCSs and maternal blood lead levels as important socioenvironmental risk factors for NDDs. To fill the existing knowledge gap, this study sought to (1) assess maternal prenatal NCSs and the range and the average of the blood lead levels in pregnant women in Suriname and to (2) analyze the single and cumulative effect of NCSs and lead exposure on their children’s neurodevelopment using the BSID III. The presence of multiple stressors more realistically depicts real-world exposure. With this cumulative model, the overall impact of multiple stressors (nonchemical and chemical) on NDDs was addressed. These data are expected to inform environmental health decision making and policy development to promote public health efforts focused on mitigating the risk factors for NDDs and to improve mother and child health in Suriname. 

## 2. Materials and Methods

### 2.1. Study Design

Suriname, a former Dutch colony located in the northern part of South America, is an upper-middle-income country. Suriname has a multiethnic/multicultural population of 541,638 inhabitants: 27.4% Hindustani (descendants of immigrants from the Indian subcontinent), 21.7% tribal people (formerly Maroons, descendants of slaves that fled into the interior), 15.7% Creoles (descendants of slaves in the coastal area), 13.7% Javanese (descendants of immigrants from Indonesia), 13.4% mixed people, and 3.8% indigenous people (formerly Amerindians) [31]. People of tribal descent mainly live in communities in the interior of tropical rainforests, and they also migrated to areas in Paramaribo and to nearby districts such as Wanica. The remainder (4.3%) represents Chinese (descendants of immigrants from China), Caucasians, and other ethnic groups such as Brazilians and Lebanese [31]. Most of the Surinamese population lives in the capital, Paramaribo, and the neighboring district of Wanica (66.3%) [31]. 

The maternal and child health threat is one of the most prominent health threats facing Surinamese women. Specifically, one in five pregnancies ends in at least one adverse birth outcome such as preterm birth or low birth weight [32]. In 2016, Suriname had 7207 cases of NDDs/100,000 children younger than 5 years of age [3]. In addition, rates of maternal prenatal NCSs in the Caribbean Consortium for Research in Environmental and Occupational Health (CCREOH) cohort have been assessed and published [33,34]. These studies reported a prevalence of high perceived stress for the CCREOH cohort ranging from 25% to 27% [34] and, for the Paramaribo subcohort, of 30% [33]. The prevalence of probable depression in the CCREOH cohort ranged from 18% to 22% [34]. 

The CCREOH MeKiTamara study is a prospective environmental epidemiologic cohort study in Suriname that addressed the impact of nonchemical and chemical environmental exposure in mother–child dyads in Suriname [35]. Pregnant women were recruited in the first or early second trimester of pregnancy from four hospitals (Academic Hospital Paramaribo, Diakonessen Hospital, s’ Lands Hospital, and Sint Vincentius Hospital), prenatal clinics and midwife facilities of the Regional Health Services, and multiple health care clinics of the Medical Mission Primary Health Care Suriname during the period of December 2016 to July 2019. Women were eligible if they were 16 years or older; spoke Dutch, Saramaccan, Sranan Tongo, or Trio; had a singleton gestation; were planning to give birth at one of the study sites; and provided written informed consent. Additional assent was obtained for participants aged 16 and 17 years of age. 

### 2.2. Study Population and Procedures

The current study reported on a subset of the larger MeKiTamara CCREOH study (*n* = 1190) and was based on 363 of the 666 pregnant women residing in two urban districts of Suriname: Paramaribo and Wanica (Figure 1). These are industrialized districts with high concentrations of historic buildings and high motor traffic density. Women were recruited prenatally, and questionnaires on maternal demographics and validated questionnaires assessing prenatal NCSs (Cohen’s Perceived Stress Scale (CPSS) [36] and the Edinburgh Postnatal Depression Scale (EPDS)) [37] were administered by trained recruiters using encrypted iPads through face-to-face interviews. If a participant was unable to read, the recruiter read the questions in the local language. Data were uploaded and managed using Research Electronic Data Capture (REDCap). In addition to the questionnaires, a venous blood sample was drawn from the participants during initial recruitment. Precoded inclusion kits were available at each inclusion site, containing the study information forms, informed consent forms, questionnaires, and blood tubes.

### 2.3. Maternal Perceived Stress

Maternal perceived stress was measured with the CPSS. This 10-item questionnaire asks about one’s feelings and thoughts in situations defined as uncontrollable and unpredictable during the last month. The CPSS was validated in multiethnic populations and is a reliable instrument for measuring perceived stress [36]. The 75th percentile (score of ≥20) was used as the cut-off level for characterizing “high perceived stress’” and “low perceived stress” (score of <20) in the participants [36]. The Cronbach’s alpha of the CPSS was between 0.84 and 0.86 [38]. Solivan et al. [39] found in their study a positive correlation between the CPSS and the EPDS.

### 2.4. Maternal Probable Depression

Probable depression during pregnancy was measured using the EPDS, a widely used 10-item self-report questionnaire that was previously validated for use in prenatal populations to identify levels of probable depression [37,40]. Women were asked to rate their mood, feelings, and thoughts experienced and or expressed in the previous seven days. Probable depression was first analyzed in four categories: (1) no or minimal probable depression (EPDS score of ≤6), (2) mild probable depression (EPDS score of 7–13), (3) moderate probable depression (EPDS score of 14–19), and (4) severe probable depression (EPDS score of 20–30) [41]. Subsequently, due to small sample sizes, the moderate and severe categories were combined, resulting in a three-level characterization with categories of (1) no or minimal (score of 0–6), (2) mild (score of 7–13), and (3) moderate or severe (score of 14–30) probable depression. 

### 2.5. Prenatal Blood Lead Levels

Whole blood was collected from maternal participants through venipuncture in 10 mL trace element vacutainers with K_2_EDTA anticoagulant. Samples were stored at 4 °C at the collection site for no more than 48 h prior to delivery to the Academic Hospital Paramaribo’s clinical laboratory. Once at the Academic Hospital Paramaribo, the blood was processed and transferred into lead-free cryovials at −80 °C. Frozen samples were shipped for analysis to the Wisconsin State Laboratory of Hygiene, Trace Element Research Laboratory, using established chain-of-custody protocols (Madison, WI, USA). Lead concentrations were determined in a 500 µL volume of whole blood using magnetic sector field inductively coupled plasma mass spectrometry (SF-ICPMS). SF-ICPMS can detect the presence of blood lead at levels well below the BLRV [42]. Prenatal exposure to lead was first examined as a continuous variable and, subsequently, as a categorical variable with the current USCDC BLRV (3.5 µg/dL) [19] as the cut-off level. From the final sample (*n* = 363), blood lead levels were available for 153 pregnant women. 

### 2.6. Maternal Sociodemographics

Maternal age, parity, ethnic background, educational level, household income, household size, and marital status were explored as covariates. Initially, age was explored as a continuous variable and then, subsequently, as a categorical variable (16–19 vs. 20–24 vs. 25–29 vs. 30–34 vs. 35+ years old). Parity, based on the number of previous live births, was first categorized as no vs. one vs. two vs. three vs. four or more and then, in the subsequent analyses, was dichotomized into 0–3 vs. 4+ previous live births based on the association with the outcome. Ethnic background was based on self-report and was categorized as Creole, Hindustani, indigenous, Javanese, tribal, and mixed. Educational level, based on the participants’ report of their highest grade and degree completed, was initially explored as not educated/primary education vs. lower secondary/vocational vs. upper secondary/vocational vs. tertiary. Subsequently, it was categorized as lower education (not educated/primary/lower secondary/lower vocational) vs. higher education (upper secondary/upper vocational/tertiary) due to small sample sizes. Monthly household income in Surinamese dollars (SRD) was analyzed in four subgroups (<800 vs. 800–1499 vs. 1500–2999 vs. 3000+ SRD (USD 210)). Household size, including the participant, was analyzed as a dichotomous variable (<3 vs. 3+ persons in household). Marital status was dichotomized as married or living with partner vs. unmarried or not living with partner.

### 2.7. Toddlers’ Sex

Sex was dichotomized as biologically male vs. female. 

### 2.8. Neurodevelopment Outcomes 

The BSID III was used to determine neurodevelopment, which was completed when the children were between 12 and 24 months of age. The BSID III is a widely used, individually administered, age-adjusted instrument designed to measure cognitive, motor, language, social-emotional, and adaptive behavior developmental functioning of infants and toddlers from 1 month to 42 months of age and to identify possible neurodevelopmental delay or disabilities [43,44]. The BSID III has recently been validated for use in Suriname [45]. The BSID III consists of a series of developmental playing tasks used to assess cognitive, motor, and language development of infants and toddlers [44]. Assessment of the social-emotional domain was conducted using primary caregiver responses to a questionnaire. Infants born premature at a gestational age less than 33 completed weeks or with a birth weight less than 2000 grams or infants and toddlers with a significant medical or neurological condition, including Down’s syndrome, autism, hydrocephalus, cerebral palsy, or significant visual or hearing impairment preventing neurocognitive testing, were excluded. Testing was performed by psychologists or trained researchers who were blind to the data on maternal prenatal NCSs and blood lead levels. Consistent with BSID III administration instructions, chronological age was calculated and utilized to determine the age-specific start point before administration. Administration and interpretation of the BSID III was based on the instructions in the technical manual [44]. After the assessment, the examiner summarized the raw score of each domain, converted them to scaled scores, and rechecked the scores before entering them in the database. The primary outcome measures were the scaled scores of the cognitive, gross and fine motor, receptive and expressive language, and social-emotional development domains, ranging from 1 to 19 with a mean of 10 and a standard deviation of 3. The scaled scores were analyzed as continuous variables. In total, 363 valid BSID III assessments between 12–24 months of age were analyzed (Figure 1).

### 2.9. Statistical Analyses

Missing data on neurodevelopment and blood lead levels were tested to determine if they were missing at random. Normality for continuous variables was tested using the Shapiro–Wilk test. Descriptive analysis of the variables was performed and presented as means with standard deviations for normally distributed continuous variables, as median with interquartile range (IQR) for skewed distributed continuous variables, or as proportions for categorical variables. The relationship between the categorical covariates was tested through poly/tetrachoric correlation. The general linear model used to examine the independent effects of maternal NCSs and blood lead levels on the Bayley scaled scores was the ordinary least squares regression. A cumulative risk index with lead, stress, and probable depression as continuous variables was developed using weighted quantile sum (WQS) regression to create an index of combined NCSs and lead exposure. The WQS index was estimated by ranking exposure concentrations in quartiles and weights for each type of exposure, which were assigned based on their contribution to the cumulative risk index. General linear models were used to assess whether this index was independently associated with each of the BSID III scaled scores. Selection of covariates to be used in the regression model was based on significance of the bivariate analysis as well as empirical data. For the descriptive analyses, statistical significance was assumed at the level of *p* < 0.05. For all other analyses, statistical significance was assumed at the level of *p* < 0.01 due to multiple testing, as we tested six BSID III outcomes. All analyses were performed using the Statistical Package for Social Sciences (SPSS) version 27.0 for Windows (IBM Corp., released in 2020, Armonk, NY, USA).

## 3. Results

The general characteristics of the 363 mother–child dyads are presented in Table 1. The toddlers’ median (IQR) BSID III scaled scores were 11.0 (9–13) for cognitive, 11.0 (9–13) for fine and 10.0 (9–12) for gross motor, 9.0 (8–11) for receptive and 9.0 (8–11) for expressive communication, and 7.0 (7–9) for social-emotional. The maternal prenatal median (IQR) perceived stress and probable depression scores were, respectively, 16.0 (12–19) and 7.0 (4–10); 24.6% had high perceived stress and 12.0% had moderate to severe probable depression. The distribution of the maternal perceived stress scores (*p* = 0.004) and depression scores (*p* = 0.038) were statistically significantly different between participants with BSID III testing and nonparticipants without BSID III testing. The maternal prenatal median (IQR) perceived stress and probable depression scores were, respectively, 16.0 (12–19) and 7.0 (4–10) for participants with BSID III testing and were, respectively, 18.0 (14–20) and 8.0 (5–12) for nonparticipants without BSID III testing (Appendix A). There were no other differences. Out of 584 live births, 221 toddlers did not have a neurodevelopment test due to (1) withdrawal or being lost to follow-up during the time between maternal recruitment and the time Bayley was to be administered, (2) the exclusion of children born preterm (<33 weeks of gestation) or with a birthweight lower than 2000 grams, or (3) invalid Bayley testing (*n* = 17). In Appendix A, we compared the maternal demographics among participants with and without Bayley testing. 

Out of 584 live births, 204 toddlers did not complete a neurodevelopment test at the time of this study, and 17 toddlers had invalid test results (Figure 1). Median (IQR) blood lead levels were 1.7 µg/dL (1.1–2.6); 11.1% were above CDC action levels (5 μg/dL). When comparing participants with BSID III scaled scores with and without blood lead levels, we found that the distribution of household income was statistically significantly different for these two categories of participants (*p* = 0.04). ( Appendix A). There were no other key demographic differences between participants with and without blood lead levels. The median (IQR) maternal age at prenatal intake was 28.4 (24.7–33.0) years of age. The majority of participants had no (35%) or one (34%) previous live birth and were Creole (34%) or had a mixed (24%) ethnic background. In terms of their socioeconomic factors, the majority of mothers had a secondary educational level (67%), a household income of 3000+ SRD (41%), a household size of 3+ persons (89%), and were married or lived together (81%). The median (IQR) age of the toddlers at BSID III assessment was 16 (15–20) months; 50.4% were boys.

Appendix A describes the correlation between the maternal variables. Education was significantly positively correlated with household income (95% CI: [0.16–0.69]). Although this correlation was negligible (*r* = 0.05), there were no other statistically significant correlations between the variables. ( Appendix A).

Table 2 shows the unadjusted associations between NCSs, lead levels, and maternal sociodemographics for each of the BSID III scaled scores. Male toddlers had significantly lower receptive (8.9 ± 2.2 vs. 9.6 ± 2.1; *p* = 0.002) and expressive (8.9 ± 2.1 vs. 9.5 ± 2.1; *p* = 0.004) communication scores compared to female toddlers. In additional analyses, no significant differences due to infant sex were found for maternal perceived stress, probable depression, educational level, or blood lead levels. Compared to a higher educational level, a lower educational level was associated with significantly lower cognitive (mean 10.3 ± 2.7 vs. 11.4 ± 2.5; *p* < 0.001), fine motor (mean 10.5 ± 2.4 vs. 11.2 ± 2.4; *p* = 0.006), and receptive communication (mean 8.9 ± 2.1 vs. 9.5 ± 2.2; *p* = 0.002) scores.

Toddlers with lead levels of ≥3.5 µg/dL had significantly lower receptive (7.8 ± 2.0 vs. 9.2 ± 2.0; *p* = 0.008) and expressive (8.8 ± 2.0 vs. 9.4 ± 2.0; *p* = 0.006) communication scores compared to toddlers with lead levels of <3.5 µg/dL. Toddlers born to mothers with no or minimal probable depression had significantly higher gross and fine motor scores compared to toddlers born to mothers with mild probable depression (mean difference 0.8 ± 0.3; 95% CI [0.11–0.146]; *p* = 0.017) and severe probable depression (mean difference 1.3 ± 0.5; 95% CI [0.20–2.41]; *p* = 0.014). Prenatal perceived stress and other maternal demographics (maternal age, parity, ethnic background, household income, household size, and marital status) were not significantly associated with the BSID III scores.

Table 3 shows that after adjusting for the toddlers’ sex and for maternal age, parity, ethnic background, education, household size, and marital status, probable depression remained significantly associated with the gross motor scores (β −0.13, 95% CI: [−0.24–−0.02]) in addition to marital status (β 1.19, 95% CI: [0.10–2.28]). After adjustment for these covariates, the lead levels remained significantly associated with the receptive communication scores (β −0.26, 95% CI: [−0.49–−0.02]). Education was also positively associated with the receptive communication scores (β 0.59, 95% CI: [0.34–1.53]). The sex of the toddler remained significantly associated with expressive communication (β 0.74, 95% CI: [0.05–1.42]) after adjustment for these covariates. Combined exposure, including exposure to stress, probable depression, and lead, was measured using the cumulative risk index. The cumulative risk index assigned the highest weight to depression followed by stress and lead exposure for the cognitive subscale; for the fine motor and expressive communication subscale, the highest weight was assigned to lead exposure followed by stress and depression; for the receptive communication and social-emotional subscales, the highest weight was assigned to stress followed by lead exposure and depression. Finally, for the gross motor subscale, the highest weight was assigned to depression followed by lead exposure and stress. The cumulative risk index was significantly negatively associated with the fine motor scores after adjustment for these covariates (β −0.74, 95% CI: [−1.41–−0.01]). Also, ethnic background was significantly associated with the fine motor scores (β −0.23, 95% CI: [−0.42–−0.04])

## 4. Discussion

To our knowledge, this is the first study to explore the single and combined effect of prenatal NCSs and lead exposure on the neurodevelopmental outcomes in toddlers living in Suriname. We found evidence that a cumulative risk score, based on combined prenatal exposure to maternal prenatal perceived stress, probable depression, and lead, predicted poor fine motor development based on the BSID III compared to estimates based on single exposure. In the adjusted models, the combined effect was specific to the fine motor domain of the BSID III, while maternal probable depression and lead exposure independently predicted lower gross motor and language scores, respectively. Our findings contribute to the existing scarce evidence that combined exposure to prenatal NCSs and lead affects child neurodevelopment. 

We found that approximately one out of four pregnant women in Paramaribo and Wanica self-reported high perceived stress and that one out of eight had probable moderate to severe depression. Our results were consistent with previous research that found a prevalence of maternal prenatal stress ranging from 11% to 29% in developing countries [34,46,47,48]. Furthermore, our rates of depression were also consistent with a meta-analysis of antepartum depression in LMICs in which rates ranged from 22.7% to 34%. [49]. We previously reported a slightly higher prevalence of maternal prenatal high perceived stress [33]; however, this was in a different subset of the cohort and included participants from other districts in addition to Paramaribo and Wanica. The current study was a priori limited to Paramaribo and Wanica due to the focus on blood lead levels and expected regional differences in exposure to environmental sources of lead by district. Gokoel et al. [34], using the EPDS, reported a prevalence of probable depression of 18% to 22% in the entire CCREOH cohort. Our finding in this subset of a lower rate of maternal probable depression was likely due to the regional restriction to the urban centers Paramaribo and Wanica. In addition, we classified probable depression into four categories, while Gokoel and colleagues classified probable depression into two categories. Findings on the prevalence of psychosocial stress vary across studies due to differences in the study design, the method, the specific type of psychosocial stress measured (i.e., perceived stress, anxiety, and depression), and the sociodemographic variables, making direct comparisons challenging. 

We found that the male toddlers had lower scores in the language domains of the BSID III through bivariate analyses. While studies do not generally report sex differences in BSID III scores, there is a substantial literature base documenting sex differences in relation to prenatal exposure [50,51]. Furthermore, there are sex-dependent differences, such as differences in brain area volume, cell number, synaptic connectivity, and neurotransmitter systems, in specific regions of the brain between males and females as well as differences in developmental trajectories by sex [52,53]. These sex differences, which could affect the susceptibility and the response of males and females to a “toxic” environment, could be an explanation for why the male toddlers performed poorer on the BSID III test compared to the female toddlers.

In our study, the mean blood lead level was 2.13; ranged from 0.39–10.02 μg/dL; and was slightly lower than that reported in a study in French Guiana, where the mean blood lead level was 3.26 μg/dL in pregnant women [54]. In the Caribbean, Grenada, and St. Vincent, the blood lead levels in pregnant women ranged from 1.17 μg/dL (Grenada) to 1.98 μg/dL (St. Vincent) [51]. Compared to U.S. and Canadian data, the blood lead levels in Caribbean women are generally higher, highlighting the need to implement surveillance programs that continuously monitor and evaluate the lead levels in the blood, especially in pregnant women, and in the environment to identify high-risk pregnant women and communities for prevention and intervention efforts in order to lower exposure and the associated future health risks. [55]

Toddlers from mothers with blood lead levels of 3.5 µg/dL and higher performed significantly poorer on the language domains of the BSID III scales in the bivariate analyses and in the fully adjusted models. Our study provided key evidence in support of the renewed recommendations that there truly is no “safe” level of lead and that blood lead levels lower than the previous USCDC BLRV of 5 μg/dL can result in negative effects on neurodevelopment. Our results were also consistent with the previously reported findings of Hu et al. [22] and Tamayo Y Ortiz et al. [14] who reported that blood lead levels of 3.5 µg/dL or higher negatively impacted language and cognitive functioning in toddlers. These studies used a previous version of the Bayley (BSID II) and the MDI scoring to assesses combined cognitive and language development, while, in the BSID III, the scoring separates the language and cognitive domains. Our findings suggested that there may be a specific effect of the blood lead level on language development and that measures that combine the language and cognitive domains may be less able to detect detrimental effects. Alternatively, as the scores in all the domains were lower, albeit not statistically significant, in the toddlers from mothers with high blood lead levels, it may be that the effect size of the blood lead level on the other domains is smaller and would, therefore, require a larger sample size to detect significant associations.

The evidence indicated associations between prenatal depression and poorer cognitive and language development [15,56]. However, in these studies, the children were 30 and 18 months of age, respectively. In our study, maternal depression was associated with the motor scores. Consistent with our results, a previous study reported an association between prenatal depression and lower scores in motor maturity in male newborns using the Brazelton Neonatal Behavior Assessment Scale [57]. Other studies did report an association with motor development in children. Servili et al. [58] found that antenatal maternal mental disorder symptoms were associated with poorer motor BSID II scores in children that were 12 months of age, but this became nonsignificant after adjusting for confounders. In their study, mental disorders included not only depression but also anxiety and somatoform disorders. Tuovinen et al. [59] reported lower motor scores in children between two and six years of age. However, they used different tools to measure depression and child neurodevelopment. While a direct comparison with our results is not feasible, this suggests that the effects of prenatal maternal depression may span the cognitive, language, and motor domains, but more research is needed. 

As prenatal depression is often associated with an increased risk of postpartum depression, this represents an alternative hypothesis for our results [60,61]. Specifically, depressed mothers may be less responsive and interactive with their children or may provide decreased opportunities to explore their environment, which are both key factors related to the early development of motor skills [62,63]. Consistent with this, several studies reported an association between postpartum depression and infant motor scores. Black et al. [64] observed lower BSID II motor scores in infants whose mother were depressed postpartum. Ali et al. [65] and Hadley et al. [66] also reported lower gross and fine motor scores in the infants and toddlers of mothers who were depressed postpartum. Given that prenatal depression is a substantial risk factor for postpartum depression, it may be that our findings are reflective of the combined exposure to prenatal and postnatal depression; unfortunately, postnatal depression data was not collected. Future studies should carefully characterize the perinatal environment, as interventions for both prenatal and postpartum depression exist and may be key prevention approaches for delayed motor development in toddlers whose mothers have elevated depressive symptoms.

In addition to the blood lead level and maternal probable depression, we also found that maternal education was a significant predictor for the BSID III scaled scores. Specifically, a lower maternal educational attainment was significantly associated with lower cognitive, fine motor, and receptive communication scores in the bivariate analyses. When controlling for maternal stress, depression, and lead exposure, maternal education only remained associated with receptive communication. Our results were in line with other study results that found that maternal education is a predictor of the cognitive and language BSID III scaled scores in toddlers [67,68]. There is inconclusive evidence of the impact of maternal education on motor development, and, while some studies report an association [69,70,71], others found no associations [72,73]. Most of these studies used different measures to assess neurodevelopment, and, as such, direct comparisons should be done with caution. 

In the adjusted models, we found that prenatal exposure to probable depression or lead was significantly associated with the gross motor and receptive communication scores, respectively. Furthermore, cumulative exposure to maternal perceived stress, probable depression, and lead, even after accounting for key covariates, significantly predicted the lower fine motor BSID III scaled scores, suggesting the interactive effects of NCSs and lead exposure. The regression coefficient for combined exposure to NCSs and lead in the adjusted model was larger compared to the regression coefficient for single exposure to probable depression and lead, which suggested a greater adverse effect on the toddlers’ neurodevelopment with exposure to both, prenatal NCSs and lead, compared to the toddlers with single prenatal NCS or lead exposure. However, it is unusual that the cumulative model resulted in domain effects distinct from either of the single exposure models. One potential explanation is that maternal demographic factors such as education could have confounded the cumulative effects on the BSID III scaled scores. In the single exposure model, the effect of lead exposure on receptive communication may have been potentiated by the added impact of low educational attainment. Furthermore, marital status may have influenced the effect of probable depression on the BSID III scores. In addition, based on the literature, which suggests that stress may modify the maternal biological processes related to the other two exposures, leading to an increased risk to the fetus, stress was included in the cumulative risk exposure model. Although stress itself, in our analyses, was not an independent predictor of the BSID III outcomes, it may have attenuated the effect of the combined exposure on the BSID III scores. To our knowledge, no previous studies have examined the combined effect of prenatal stress, probable depression, and lead exposure on toddlers’ developmental outcomes, and, certainly, our data suggested the need for additional studies. 

Our results have profound population-level implications for maternal–child health in Suriname. One of ten pregnant women in this study had all three risk factors: a blood lead level above the current CDC actionable level, high perceived stress, and probable depression. Considering that an estimated 8000 live births occur each year in Paramaribo and Wanica [74], an estimation of 800 mother–infant dyads is likely at high risk for both the maternal perinatal outcomes associated with maternal NCSs and lead exposure as well as poorer neurodevelopmental trajectories for the infants and toddlers. The results of this study, while raising many additional questions, highlighted the importance of examining prenatal coexposure to NCSs and CSs due to their high rate of cooccurrence and their likely interactive effects on fetal development. Healthcare workers in Suriname need to consider including NCS and CS screening in routine prenatal consultations to accurately identify the children at risk for adverse neurobehavioral disorders. Addressing the sources of prenatal NCSs and CSs is one needed public health approach to addressing the high rates of perinatal complications for both the infant and mother in Suriname.

### 4.1. Limitations

Both maternal probable depression and perceived stress were obtained through self-report questionnaires. This approach may be associated with retrospective recall biases or socially desirable responses, resulting in an underestimation of the prevalence of stress and depression in this group of women. Furthermore, we did not assess postpartum depression or the maternal use of antidepressants. Although NCSs were assessed at two time points during pregnancy, this study utilized only the first assessment, limiting the ability to evaluate the change or cumulative maternal stress and depression across the course of pregnancy.

The blood lead level was also assessed once during pregnancy at the time of this study, precluding the evaluation of the change in the blood lead level during pregnancy. As the blood lead level is predicted to follow a U-shaped curve during gestation, a single assessment of the blood lead level early in pregnancy may not fully reflect its cumulative exposure to the fetus and may not be sufficient for establishing the full nature of risk to fetal neurodevelopment [75]. We were not able to control for postnatal lead exposure, including lead exposure due to breastfeeding, which may have influenced the effects of prenatal lead exposure. It is also known that other dietary minerals may influence lead dynamics; however, we did not assess dietary minerals, such as calcium and iron, in this study [75,76,77,78,79,80,81,82]. In addition, maternal anemia during pregnancy is an independent contributor to poor neurodevelopment that was also not measured in this study [83,84]. 

Our final analytic sample with both the blood lead level and BSID III data represented only a subset of the mothers enrolled in the study, and there were some demographic differences. Specifically, the mothers included in this analysis with BSID III data had significantly lower perceived stress than mothers whose children did not complete the BSID III. While there were differences in the level of depression, the mean differences (7 vs. 8) are unlikely to represent a significant difference in depressive symptoms. Because of budgetary constraints, we could not measure the blood lead level of all the participants. Maternal blood samples were collected and analyzed randomly from the mothers recruited into the CCREOH cohort. Consequently, the blood lead levels were not available for all the mothers in this subcohort, resulting in some children with BSID III data but no maternal blood lead levels. As such, future studies with larger and more complete samples are needed.

### 4.2. Strengths

Our study is the first study in Suriname to assess the single and combined effect of prenatal NCSs and lead exposure during pregnancy on child neurodevelopment in a reasonably sized population-based study using the BSID III, which was recently validated in Suriname [45]. Research combining these types of exposure in humans remains scarce [85]. Prenatal exposure to these social-environmental factors in developing countries may differ from those in industrialized countries, and may be much harsher and deserve to be studied. Many women do not know that they are (were) exposed to lead or that they have mental health disorders. They go unnoticed and undiagnosed. Addressing persistent disparities in perinatal and long-term neurodevelopmental trajectories across the globe requires culturally responsive assessments and evaluations in order to define the sources of the factors that threaten healthy child development. Understanding the specific environmental and psychosocial challenges to child well-being in each country is a key step toward the design and implementation of culturally relevant prevention and intervention efforts.

## 5. Conclusions 

This research, conducted on a subcohort of the CCREOH environmental epidemiologic study, confirmed that (1) the unborn children in Paramaribo and Wanica, Suriname, were exposed to NCSs and lead and that (2) NCSs and lead exposure, independently and in combination, had negative consequences on their neurodevelopment. 

These findings set a precedent to implement psychosocial stress screening, including social services for NCSs, in a culturally responsive manner and targeted lead exposure screening in the primary care setting for pregnant women and children at specific ages. In Suriname, there is a need for interventions in order to mitigate or lower these types of exposure to improve maternal and child perinatal and long-term outcomes. Moreover, from a public health perspective, it is important to identify pregnant women at high risk for exposure to both NCSs and lead given their established ties to elevated health risks for their offspring. 

We should continue to monitor both NCSs and lead exposure in these women, as many will have more children, and there is also a need to monitor these children for additional exposure to both lead and NCSs in the postnatal environment. The prospective observation of these children should continue across their development to determine the full impact of these types of prenatal exposure on neurodevelopment. The deleterious effects of NCSs and lead exposure on child neurodevelopment may develop over the course of early childhood and have the potential to become more clinically relevant as the child enters school and as the neurocognitive demands placed on children increases. The longitudinal assessment of all the neurodevelopmental domains with precise measurements that do not solely rely on a parental report will likely provide the most informative data on the long-term consequences of cumulative exposure to prenatal and postnatal chemical and nonchemical factors that are known to be detrimental to healthy child development.

## Figures and Tables

**Figure 1 children-10-00287-f001:**
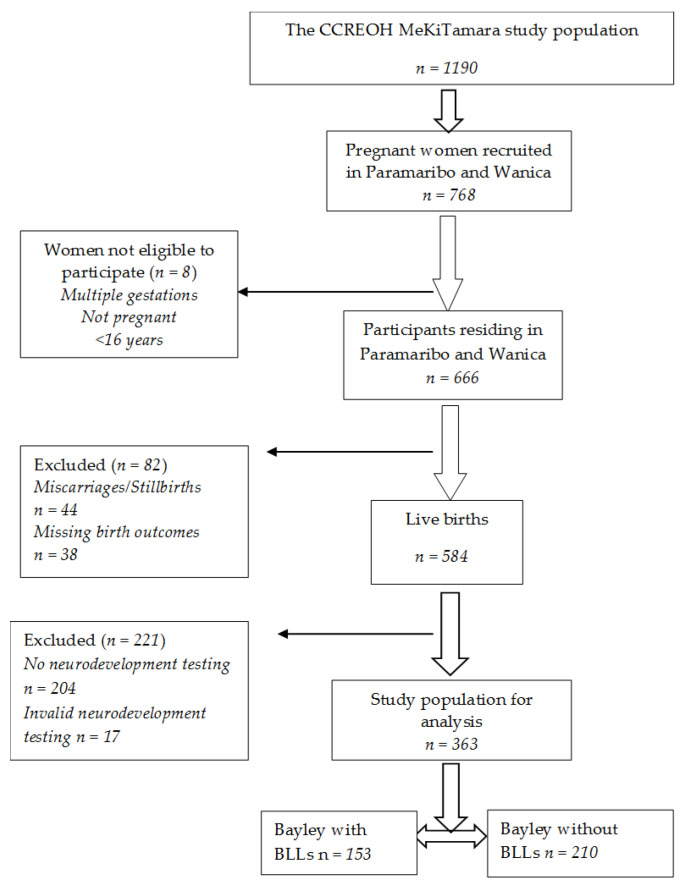
Flowchart of the study population (BLLs is blood lead levels).

**Table 1 children-10-00287-t001:** General characteristics of the CCREOH subcohort Paramaribo and Wanica (*n* = 363).

Variables	*n*	%	% Missing
Toddlers’ characteristics			
Sex			0.0%
Boys	183	50.4%	
Girls	180	49.6%	
Age (in months)	16 (15–20)		
Median (IQR)			
BSID III scaled scores. Median (IQR)			0.0%
Cognitive	11 (9–13)		
Fine motor	11 (9–13)		
Gross motor	10 (9–12)		
Receptive	9 (8–11)		
Expressive	9 (8–11)		
Social-emotional	7 (7–9)		
Maternal characteristics			
Blood lead levels			57.9%
Median (IQR)	1.7 (1.1–2.6)		
<3.5 µg/dL	136	88.9%	
≥3.5 µg/dL	17	11.1%	
Cohen’s Perceived Stress Scores			6.1%
Median (IQR)	16 (12–19)		
0–19 low–normal	261	75.4%	
20–40 high	85	24.6%	
Edinburgh Depression Scores			6.3%
Median (IQR)	7 (4–10)		
0–6 no or minimal	157	46.2%	
7–13 mild	142	41.8%	
14–19 moderate	30	8.8%	
20–30 severe	11	3.2%	
Age (in years)			0.0%
Median (IQR)	28.4 (24.7–33.0)		
16–19	32	8.8%	
20–24	65	17.9%	
25–29	117	32.2%	
30–34	82	22.6%	
35+	67	18.5%	
Parity			0.3%
0 previous live births	125	34.5%	
1 previous live birth	113	31.2%	
2 previous live births	60	16.6%	
3 previous live births	31	8.6%	
4+ previous live births	33	9.1%	
Ethnic background			0.0%
Creole	125	34.4%	
Hindustani	64	17.6%	
Indigenous	5	1.4%	
Javanese	14	3.9%	
Tribal	68	18.7%	
Mixed	87	24.0%	
Educational level			0.0%
primary or not educated	37	10.2%	
lower secondary/vocational	124	34.2%	
upper secondary/vocational	120	33.1%	
tertiary	82	22.6%	
Household income (in SRD)			4.7%
<800	23	6.6%	
800–1499	60	17.3%	
1500–2999	113	32.7%	
3000+	150	43.4%	
Household size			0.3%
<3 persons	39	10.7%	
3+ persons	324	89.3%	
Marital status			0.0%
Married/living together	295	81.3%	
Unmarried/single	68	18.7%	

**Table 2 children-10-00287-t002:** Distribution of unadjusted BSID III scaled scores for maternal demographics as well as chemical and nonchemical stressors (*n* = 363).

	BSID III Scale Scores (Mean ± SD)					
	Cognitive	Fine Motor	Gross Motor	Receptive Communication	Expressive Communication	Social-Emotional
Sex of infant						
Boys (*n* = 183)	10.9 ± 2.6	10.6 ± 2.5	10.5 ± 2.6	8.9 ± 2.2	8.9 ± 2.1	8.3 ± 2.7
Girls (*n* = 180)	10.9 ± 2.7	11.1 ± 2.4	10.2 ± 2.7	9.6 ± 2.1	9.5 ± 2.1	8.0 ± 2.6
*p*-value	0.989	0.033	0.302	0.002	0.004	0.283
Blood lead levels						
<3.5 µg/dL (*n* = 136)	11.0 ± 2.8	11.0 ± 2.6	10.2 ± 2.6	9.2 ± 2.0	9.4 ± 2.0	8.5 ± 2.6
≥3.5 µg/dL (*n* = 17)	10.5 ± 2.8	10.9 ± 2.6	9.9 ± 1.8	7.8 ± 2.0	8.8 ± 2.0	8.0 ± 2.9
*p*-value	0.453	0.885	0.617	0.008	0.006	0.416
Cohen’s Perceived Stress Scores						
0–19 low–normal (*n* = 261)	11.1 ± 2.7	10.9 ± 2.4	10.5 ± 2.6	9.4 ± 2.2	9.2 ± 2.1	8.0 ± 2.3
20–40 high (*n* = 85)	10.3 ± 2.6	10.6 ± 2.6	10.0 ± 2.7	8.8 ± 2.1	9.3 ± 2.2	8.7 ± 3.4
*p*-value	0.025	0.249	0.122	0.058	0.668	0.019
Edinburgh Depression Scores						
0–6 no or minimal (*n* = 157)	11.2 ± 2.7	11.2 ± 2.4	10.8 ± 2.8	9.4 ± 2.3	9.2 ± 2.1	8.1 ± 2.3
7–13 mild (*n* = 142)	10.7 ± 2.7	10.5 ± 2.5	10.2 ± 2.5	9.1 ± 2.1	9.1 ± 2.3	8.0 ± 2.5
14–30 moderate and severe (*n* = 41)	10.3 ± 2.7	10.3 ± 2.5	9.5 ± 2.5	9.0 ± 2.1	9.3 ± 1.9	8.8 ± 4.0
*p*-value	0.114	0.009	0.009	0.383	0.816	0.207
Age (in years)						
16–19 (*n* = 32)	10.9 ± 2.4	10.8 ± 2.5	10.5 ± 2.6	9.5 ± 2.5	9.0 ± 2.1	8.1 ± 3.3
20–24 (*n* = 65)	10.6 ± 2.4	10.5 ± 2.4	11.0 ± 3.2	9.7 ± 2.1	9.5 ± 2.2	8.2 ± 2.5
25–29 (*n* = 117)	11.2 ± 3.0	11.1 ± 2.3	10.2 ± 2.5	9.0 ± 2.2	9.0 ± 2.0	8.1 ± 2.6
30–34 (*n* = 82)	10.9 ± 2.5	10.7 ± 2.4	10.2 ± 2.3	9.1 ± 2.2	9.5 ± 2.5	8.1 ± 2.8
35+ (*n* = 67)	10.8 ± 2.5	11.0 ± 2.6	10.3 ± 2.6	9.3 ± 2.0	9.0 ± 1.8	8.1 ± 2.2
*p*-value	0.725	0.486	0.310	0.250	0.370	0.998
Parity						
0–3 previous live births (*n* = 329)	11.0 ± 2.6	10.9 ± 2.5	10.4 ± 2.7	9.3 ± 2.1	9.2 ± 2.1	8.1 ± 2.6
4+ previous live births (*n* = 33)	10.4 ± 3.0	10.5 ± 2.2	10.0 ± 1.7	8.7 ± 2.2	8.9 ± 2.2	8.7 ± 3.0
*p*-value	0.225	0.392	0.427	0.150	0.439	0.182
Ethnic background						
Creole (*n* = 125)	11.2 ± 2.5	11.2 ± 2.3	10.7 ± 2.7	9.3 ± 2.1	9.2 ± 2.1	7.9 ± 2.5
Hindustani (*n* = 64)	10.9 ± 2.8	10.9 ± 2.4	9.9 ± 2.7	9.6 ± 2.1	9.6 ± 2.3	8.3 ± 2.5
Indigenous (*n* = 5)	10.0 ± 1.6	12.8 ± 2.0	10.4 ± 1.5	9.2 ± 1.3	9.0 ± 1.2	7.2 ± 3.1
Javanese (*n* = 14)	10.4 ± 2.3	10.7 ± 2.3	9.9 ± 3.0	9.8 ± 1.9	9.7 ± 2.0	7.7 ± 2.1
Tribal (*n* = 68)	10.4 ± 2.7	10.1 ± 2.3	10.2 ± 2.2	8.7 ± 2.1	8.8 ± 1.9	8.4 ± 3.5
Mixed (*n* = 87)	10.9 ± 2.7	10.8 ± 2.7	10.5 ± 2.9	9.3 ± 2.3	9.2 ± 2.3	8.3 ± 2.0
*p*-value	0.353	0.028	0.441	0.267	0.391	0.652
Educational level						
lower (*n* = 161)	10.3 ± 2.7	10.5 ± 2.4	10.4 ± 2.5	8.9 ± 2.1	9.0 ± 2.0	8.2 ± 3.1
higher (*n* = 202)	11.4 ± 2.5	11.2 ± 2.4	10.6 ± 2.8	9.5 ± 2.2	9.4 ± 2.2	8.1 ± 2.2
*p*-value	< 0.001	0.006	0.140	0.002	0.110	0.650
Household income (in SRD)						
<800 (*n* = 23)	11.4 ± 3.1	11.2 ± 2.3	10.3 ± 1.6	9.3 ± 1.7	9.2 ± 1.9	8.5 ± 3.4
800–1499 (*n* = 60)	10.2 ± 2.7	10.1 ± 2.2	9.5 ± 2.3	8.8 ± 1.8	8.9 ± 2.2	8.4 ± 3.3
1500–2999 (*n* = 113)	10.7 ± 2.7	10.8 ± 2.6	10.6 ± 3.0	9.4 ± 2.3	9.1 ± 1.9	7.7 ± 2.4
3000+ (*n* = 150)	11.4 ± 2.5	11.3 ± 2.4	10.6 ± 2.6	9.4 ± 2.1	9.5 ± 2.3	8.3 ± 2.3
*p*-value	0.019	0.010	0.050	0.223	0.154	0.209
Household size						
<3 persons (*n* = 39)	11.0 ± 2.6	11.2 ± 2.6	10.1 ± 2.5	9.1 ± 2.3	9.2 ± 2.0	7.7 ± 2.0
3+ persons (*n* = 324)	10.9 ± 2.7	10.8 ± 2.4	10.4 ± 2.7	9.3 ± 2.1	9.2 ± 2.1	8.2 ± 2.7
*p*-value	0.860	0.425	0.542	0.681	0.856	0.356
Marital status						
Married/living together (*n* = 295)	11.0 ± 2.7	10.9 ± 2.4	10.3 ± 2.7	9.2 ± 2.2	9.2 ± 2.2	8.0 ± 2.5
Unmarried/single (*n* = 68)	10.7 ± 2.6	10.8 ± 2.4	10.7± 2.6	9.2 ± 2.2	9.1 ± 2.0	8.4 ± 3.0
*p*-value	0.433	0.681	0.227	0.896	0.732	0.317

Bonferroni correction was applied because of multiple testing, and, therefore, a *p*-value of <0.01 was used for statistical significance.

**Table 3 children-10-00287-t003:** Adjusted linear regression models for single and cumulative exposure to lead, perceived stress, and probable depression based on BSID III scaled scores.

	β [95% CI: LB-UB] for BSID III Scaled Scores					
	Cognitive	Fine Motor	Gross Motor	Receptive Communication	Expressive Communication	Social-Emotional
Model 1:						
Single lead exposure	−0.08 [-0.40–0.24]	−0.08 [−0.37–0.20]	0.00 [−0.30–0.29]	−0.26 [−0.49–−0.02]	−0.19 [−0.42–0.04]	−0.12 [−0.42–0.19]
Single stress exposure	−0.01 [−0.12–0.10]	−0.04 [−0.14–0.05]	0.04 [−0.06–0.14]	0.00 [−0.08–0.08]	0.02 [−0.06–0.10]	0.06 [−0.04–0.17]
Single probable depression exposure	−0.08 [−0.21–0.04]	−0.08 [−0.19–0.02]	−0.13 [−0.24–−0.02]	−0.05 [−0.14–0.04]	−0.02 [−0.11–0.07]	0.00 [−0.12–0.12]
Model 2:						
Cumulative exposure risk index #	−0.44 [−0.95–0.06]	−0.74 [−1.41–−0.01]	−0.55 [−1.23–0.12]	−0.33 [−0.81–0.14]	−0.30 [−0.84–0.25]	0.30 [−0.34–0.94]

Abbreviations: β (beta coefficient), LB (lower bound), and UB (upper bound). # Cumulative exposure risk index of lead, stress, and probable depression. Models are adjusted for the following covariates: toddlers’ sex and maternal age, parity, ethnic background, education, household size, and marital status. Because of multiple testing, a *p*-value of ≤0.01 was used for statistical significance.

## Data Availability

The data that support the findings of this study are available upon reasonable request from the corresponding author.

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
