# Peer review of "The Single and Combined Effects of Prenatal Nonchemical Stressors and Lead Exposure on Neurodevelopmental Outcomes in Toddlers: Results from the CCREOH Environmental Epidemiologic Study in Suriname"

_children, 2023, doi:10.3390/children10020287_

Round 1

Reviewer 1 Report

This is a relevant topic. The manuscript is interesting and easy to follow. The methodology is correct. Nevertheless, in the statistical analysis, I have a couple of suggestions:

1. Given the participants were recruited from different centers, I would suggest to explore any potential difference among groups. If groups look like similar, the analysis is fine; otherwise, I would think the models should be adjusted for the center the are coming

2. In relation to the models: the authors say they used "general linear models"; my point is that general linear model is the name of a broad variety of models. Please, could you be more specific?

Reviewer 2 Report

The authors measured prenatal lead exposure and psychosocial health and investigated whether this influenced child neurodevelopment at age 1 and 2 years. 

They study the factors individually and they studied the combined effect. This is interesting and especially novel for Surinam data. I have some comments with regard to clear presentation, rationale and interpreting the results.

Abstract:

* Line 32: Please use scales instead of scaled

* Line 34-35: I do not understand this: probable depression or lead, prenatal exposure to a single stressor - probable depression or lead. --> why is that double?

* For all sentences with 'lead': please add 'exposure'.

Introduction:

* Please Use μg consequently instead of ug in some cases (e.g. line 84)

* Line 92: do you have a reference supporting the first sentence?

* Line 111-130: this paragraph has no connection with the rest of the introduction and it does not really fit here. I suggest to include it in de population description.

* I suggest to not use the abbreviation BLL but write it out. You can also use maternal lead exposure.

Methods:

* Line 168: I do not understand the difference between the bigger study and the subset. Was the recruitment different (line 168)? Or, as I see in figure 1, is it that you started with women from the bigger study that lived in a certain region? In that case, include the 1190 as the first box in figure 1.

* Line 216: why were lead blood levels available for only 153 women, please describe what happened to the rest.

results:

* Line 293: I do not understand this, the missing rate of Bayley scale testing is 0% right? I now see in the table S1 that it is the ones that were excluded. It would be good to add some more explanation here.

* I am a bit surprised that there are no differences between the participants with and without a blood lead level and it makes me curious to the reason why some have this datapoint and some haven't (as questioned earlier in the review). Would it also be possible to include a table like S1 for these two groups in the supplement?

* Bayley and BSIDIII are both used, please use 1 consequently.

* Please explicitly mention the basis of the cutt-off of 3.5 μg/dl lead level in the methods section (the blood lead reference value, with reference).

* I am a bit confused regarding table 2 and 3. I suggest to only include univariate linear regression results for demographics in table 2. Move the lead exposure and psychosocial factors to table 3 and include both unadjusted and adjusted results together. Then you can easily see the differences. You can also include the multiple linear regression in table 3.

* Have you tested interaction between non-chemical and chemical factors, as you describe in the introduction?

Discussion:

* Line 364: 'some domains' is mentioned, but in fact it is only one domain, so this is a bit too strongly formulated. Please change it into the specific domain.

* Line 409: I do not understand this, you have found an association for <3.5 μg/dl, so you cannot conclude about low levels right? It could be an idea to provide a graph with the lead concentrations and the outcome scores.

* Line 444: and what did they find?

* Line 448: and how would that affect your results? And what about a woman without depression prenatally, developing post partum depression? 

* Line 458-461: second time 'remained' can be deleted, missing 'a' before 'predictor, and 'other' needs an extra 's': 'others'

* Line 468: is this interaction or an additive effect? 

* Line 526: 'BLL were not available for all mothers': I don't understand that, are you referring to a subset? If so, please mention that.

* Line 553: what kind of treatment is there for high lead exposure? Maybe high risk on lead exposure is not fitting in this sentence, and it concerns psychosocial health here. The next sentence is also mixed up I think: you mean CS instead of NCS.

Round 2

Reviewer 2 Report

Thank you for responding to the comments. I my view, the comments were sufficiently addressed.